# Review on Current Advancements in Facilitation of Burn Wound Healing

**DOI:** 10.3390/bioengineering12040428

**Published:** 2025-04-18

**Authors:** Wing Sum Siu, Hui Ma, Ping Chung Leung

**Affiliations:** Institute of Chinese Medicine, The Chinese University of Hong Kong, Shatin, New Territories, Hong Kong SAR, China; huima@cuhk.edu.hk (H.M.); pingcleung@cuhk.edu.hk (P.C.L.)

**Keywords:** burn wound, wound healing, dressing, stem cell, extracellular vesicles, traditional medicine

## Abstract

Burns are common injuries, but their treatment remains challenging due to the complex nature of the wound healing process. Burn wounds are classified into different categories based on their size and depth. Treatment modalities vary significantly across these categories, primarily focusing on the inflammation, proliferation, and remodeling phases of burn wound healing. This review summarizes current research on various approaches to enhance burn wound recovery, including advancements in wound dressings, the use of platelet-rich plasma, stem cells, their soluble factors primarily in the form of secretomes or extracellular vesicles, and nano-technologies. Additionally, advancements in modernized traditional medicine are discussed to give a new aspect for burn wound healing. This review also summarizes the barriers in translating bench research to clinical practice in burn wound treatment methods. For an effective translation, researchers and industrial partners should work more closely, while regulatory bodies should streamline the approval procedure.

## 1. Introduction

Burn injuries are a common form of trauma. It is estimated that there are 7 to 12 million people sustaining burn injuries that require medical care each year (up to 33,000 each day) [1]. A recent epidemiological study revealed that the vast majority of burns occur in low- and middle-income countries, while children and adolescents have a high incidence of burns [2]. The World Health Organization (WHO) also reported over 195,000 deaths annually from fire-related burns alone [3].

Burn wounds can be complex to treat. Severe burn wounds may take years to heal fully. In addition, the pain associated with burns is often intolerable. Therefore, burn injuries can be classified as chronic conditions as patients often endure lifelong physical and psychological challenges. They include persistent pain, mental health issues, decreasing work capacity, and reduction in quality of life. All of them can affect morbidity and mortality. Some victims may even require long-term psychological interventions.

Consequently, the medical expenses associated with burn injuries are substantial. They encompass surgery, medication, hospitalization, and specialized medical care. Additionally, the loss of work hours further exacerbates the socioeconomic burden. In countries with a very high Human Development Index, the total hospital care costs, the cost of 1% of total body surface area burned, and the cost of hospital treatment per day can reach up to USD 125,597.86, USD 11,245.04, and USD 4125.50, respectively, per patient [4].

Most burn injury deaths are caused by complications rather than burns. Reports from even 30 years ago to now indicate that sepsis leading to multiple organ failure is the most frequent cause of death in patients with burn injuries, no matter in low- or high-income countries [5,6]. Therefore, the primary goal of burn wound management is to prevent or treat infection and other complications at the beginning when patients are admitted in an emergency. However, long-term follow-up treatments to manage the wound healing cannot be ignored. Effective treatment strategies that promote burn wound healing with minimal scarring are essential to alleviate patients’ long-term psychological distress and reduce the associated socioeconomic burden.

Current conventional clinical treatments are effective in saving lives and reducing infection rates. However, subsequent procedures aimed at promoting burn wound healing remain less than satisfactory. With advancements in biotechnology and biomaterials, numerous innovative and effective treatment modalities for burn wounds have emerged. This review aims to summarize these novel approaches. Additionally, this review illustrates existing gaps in the clinical translation of these advancements and also discusses the factors that influence their successful implementation in practice.

## 2. Methods

This review was conducted in the PubMed, Google Scholar and Medline databases using a combination of the following key words: “burn”, “wound healing” and “treatment” in all fields. The search was conducted in August 2024. Additionally, a time filter was applied to limit the publications starting from 2014. Other inclusion criteria included demonstration of method to promote burn wound healing, publication in English and full text available.

The titles and abstracts of the articles obtained in the databases were reviewed first. Then, the complete texts of the eligible articles were reviewed to see whether they fit the inclusion criteria.

The methods to promote burn wound healing were summarized into six categories: modern wound dressing, platelet-rich plasma, somatic and stem cells, soluble factors derived by stem cells, nano-technology and modernized traditional medicine.

## 3. Classification of Burn Wound

Burn wounds can be categorized based on their severity, which is determined by factors, such as the patient’s age, the percentage of total body surface area burned (%TBSA), the depth of the burn, the type of burn, and the involvement of specific body parts (Table 1).

### 3.1. Classification Based on Severity by Burn Size

Burns are classified as minor, moderate, or major (severe) based on the patient’s age and the %TBSA affected. This classification system is known as the ABA system, established by the American Burn Association [7].

Minor Burns: These involve injuries covering less than 10% of TBSA in adults, less than 5% in children and the elderly, and primarily consist of superficial burns, with less than 2% classified as full-thickness burns.

Major Burns: These are defined as injuries covering more than 20% of TBSA in adults, more than 10% in children and the elderly, or more than 5% as full-thickness burns. Additionally, significant burns to critical areas, such as the face, eyes, ears, joints, or genitalia, are also classified as major burns.

### 3.2. Classification Based on Depth of Burn

First-Degree Burns (Superficial Thickness): These burns affect only the uppermost layer of the skin (epidermis). The skin appears red, congested, and dry, with victims experiencing mild pain.

Second-Degree Burns (Partial Thickness): These burns cause partial damage to the dermis and are further classified into two categories:2A (Superficial Partial Thickness): These involve injuries to the epidermis and the upper half of the dermis. Blisters and weeping are present, accompanied by pain. Scarring may occur, and victims require dressing and wound care but typically do not need surgery.2B (Deep Partial Thickness): These burns damage the deeper layers of the dermis and are often less painful due to the partial destruction of pain receptors. The wound typically has more blisters, appears drier, and is likely to result in more scarring. Surgical intervention, such as skin grafts, is usually required.

Third-Degree Burns (Full Thickness): These burns damage the entire dermis and often do not cause pain because the nerve endings are destroyed. Such wounds require protection against infection, and most necessitate surgical management.

Fourth-Degree Burns: These extend beyond the skin to affect deeper tissues, including muscle and bone. The burned areas often appear blackened and may result in significant loss of tissue.

## 4. Different Phases in Burn Wound Healing

Burn wound repair is a complex process involving four general phases: hemostasis, inflammation, proliferation and maturation/remodeling. They occur in a temporal sequence but often overlap. The duration of healing is influenced by multiple factors, including the severity of the injury, activation of the inflammatory cascade, and nutritional supply (Figure 1).

Hemostasis: This phase occurs immediately after the injury. Vasoconstriction takes place and leads to platelet activation, aggregation, and clotting induced by collagen. Platelets, along with keratinocytes, macrophages, and fibroblasts, release growth factors, such as platelet-derived growth factor (PDGF), epidermal growth factor (EGF), and transforming growth factor-β (TGFβ). They facilitate the deposition of a fibrin clot at the injury site to serve as a provisional matrix for the subsequent stages of healing.

Inflammation: This phase begins within 24 h after the injury and can last for several days to weeks, depending on the severity of the burn. The activation of immune cells triggers the secretion of pro-inflammatory cytokines, which influence the migration of epithelial cells, endothelial cells, and fibroblasts, contributing to collagen deposition. During this phase, necrotic tissue is degraded, which initiates a cascade of signals necessary for wound repair [8].

Proliferation: This phase occurs several days after the injury and can last for weeks. It is characterized by the replacement of the provisional matrix with a connective tissue matrix. It includes the processes of fibroblast proliferation, collagen deposition, angiogenesis, granulation tissue formation (new connective tissue and microscopic blood vessels), and epithelialization (formation of new epithelium). In addition, the activation of keratinocytes and fibroblasts by various cytokines and growth factors in this phase is essential for wound healing. Keratinocytes migrate across the wound, facilitating both angiogenesis and epithelialization [9]. Endothelial cells, activated by growth factors, such as vascular endothelial growth factor (VEGF), hepatocyte growth factor (HGF), and fibroblast growth factors (FGFs), also play a crucial role in initiating angiogenesis [10]. The restoration of vascular perfusion during this phase further enhances wound healing.

Remodeling: This is the final phase of healing. In this phase, collagen and elastin are continuously deposited, and fibroblasts differentiate into myofibroblasts. These myofibroblasts adopt a contractile phenotype and contribute to wound contracture [11]. Influenced by growth factors, matrix metalloproteinases (MMPs), and tissue inhibitors of metalloproteinases (TIMPs), the granulation tissue matures and the extracellular matrix (ECM) is remodeled. These events increase the tensile strength of the wound. Finally, the apoptosis of keratinocytes and inflammatory cells marks the end of wound healing, shaping the final appearance of the wound [12].

## 5. Other Factors Affecting Burn Wound Healing

In addition to the internal factors involved in the repair process described above, several external factors influence burn wound healing (Figure 1). These factors can be categorized into local or systemic. Local factors include infection, oxygenation, ischemia, and edema. Systemic factors encompass age and gender, overall health status (such as pre-existing diseases), nutrition, obesity, personal habits (including smoking and alcohol use), and other individual circumstances [13].

Given the complexity of burn injuries, effective medical modalities that facilitate healing are essential, especially for severe burn wounds. These treatments not only alleviate the suffering of victims but also help reduce the socioeconomic burden associated with burn injuries.

Burn injuries can be caused by electrical, chemical, radiation, friction, cold, or heat sources. Notably, approximately 45% of all burn injuries were caused by a flame or flash, while scalds represented more than 30% in the US within 2019–2023 [14]. Therefore, this review focuses on reviewing the current treatment modalities that promote the healing of wounds caused by thermal burns.

## 6. Conventional Methods in Burn Wound Management

The goals of burn wound treatment are to prevent infection, remove dead tissue, control pain, minimize scarring, and restore function. Standard burn wound management includes cleaning, debridement, infection management, and dressing. These procedures aim to save lives, reduce infection rates, and prevent complications.

Conventionally, antibiotic ointments or creams are applied to second-degree burns following the cleaning and debridement of the wound site. These treatments help maintain moisture in the wound and prevent or treat infections. Silver sulfadiazine (SSD) cream is commonly used as a first-line treatment for minor burns. However, its use is limited due to potential adverse effects, including neutropenia, erythema multiforme, crystalluria, and methemoglobinemia [15,16].

For deep second-degree and third-degree burns, skin grafts are often employed. This surgical procedure involves removing dead skin and replacing it with healthy skin from another part of the body. Unfortunately, transplanted skin may not always adhere properly, necessitating additional grafts, and, thus, may increase the length of hospital stay. Factors contributing to unsuccessful grafts include infection, trauma to the graft site, and inadequate blood circulation, which can impede healing. Skin graft surgery may also lead to complications, such as bleeding, contracture, discolored or patchy skin, loss of skin sensation, increased sensitivity to pain, and chronic pain after healing.

Bandages and gauzes are the most common traditional wound dressings and they are required when ointments are used. Notwithstanding that they are highly absorbent and effective for dry to mildly exudating wounds, they need to be changed regularly. Frequent dressing changes may result in minor bleeding and significant pain to patients to, therefore, elevate their stress. Additionally, these traditional dressings typically have poor adhesion properties and may not provide adequate drainage for the wound. Therefore, there is a need to create affordable modern wound dressings.

While these conventional procedures are crucial for the initial care of burn wounds, healing can be a prolonged process. Patients may face long-term physical pain and psychological stress within the long healing period. However, clinical treatments aimed at enhancing burn wound healing remain inadequate.

## 7. Modern Approaches to Promote Burn Wound Healing

With advancements in biomedical technology over the past few decades, numerous new materials and techniques have emerged to facilitate burn wound healing. Current scientific research is continuously developing innovative treatment modalities.

### 7.1. Modern Wound Dressing

Modern wound dressings are primarily composed of synthetic polymers, although natural materials are also used. They can be categorized into three main types: interactive, advanced interactive, and bioactive dressings [17,18].

Interactive Dressings: These include semi-permeable films and foams.Advanced Interactive Dressings: These comprise hydrocolloids and hydrogels, which are highly hydrophilic macromolecular networks.Bioactive Dressings: Tissue-engineered skin equivalents fall into this category.

The benefits of these modern dressings include their versatility for application on all areas of the body, cooling and protective functions, availability in various sizes, and their ability to remove heat from the wound through convection and evaporation [19]. In recent years, many of these dressings have been enhanced with antimicrobial agents, antioxidants, anti-inflammatory compounds, analgesics, and growth factors to optimize burn wound healing [20,21,22,23].

However, although the moist environment provided by modern wound dressings is beneficial for healing, it may increase the risk of infection from microorganisms and impact the healing process [24].

### 7.2. Platelet-Rich Plasma (PRP)

PRP is a blood product derived from autologous blood, characterized by high concentrations of platelets and growth factors. The concept of PRP originated in hematology during the 1970s. Initially, it was used as a transfusion product to treat patients with thrombocytopenia [25].

PRP has proven effective in treating diabetic ulcers and soft tissue ulcers by enhancing local blood supply [26]. Additionally, it promotes cartilage tissue regeneration and reduces chondrocyte apoptosis to aid tissue repair [27]. It has also been shown to alleviate pain associated with degenerative disc disease [28].

Clinical data indicate that PRP can activate various regenerative mechanisms across different disease conditions, including hemostasis, inflammation, angiogenesis, and extracellular matrix synthesis [29]. In the 1980s, PRP was demonstrated to enhance granulation tissue formation. Also, the use of platelet releasate was proven effective in patients with severe diabetic wounds [30]. More recently, the application of PRP in burn wound healing has been reported [31].

Despite these promising findings, further advancements in basic PRP research and expanded clinical applications are necessary to fully realize its potential in clinical practice.

### 7.3. Somatic and Stem Cells

Fibroblasts and keratinocytes are the somatic cells that are commonly utilized to form products for wound and burn healing [32,33]. Fibroblasts are the main cell type in the dermis. They produce ECM components and secrete growth factors, cytokines, and matrix metalloproteinases (MMPs). MMPs are important enzymes, ensuring the formation of the ECM, as well as the proliferation and differentiation of keratinocytes. While keratinocytes are the major cell component of the epidermis, they are responsible for the formation of numerous tight intercellular junctions. They are also involved in the re-epithelialization of the wound during the wound healing. Notwithstanding their important functions in wound healing, immune rejection is commonly reported with allogeneic fibroblasts and keratinocytes [34].

Innovations in stem cell technologies offer promising therapeutic modalities for treating burn wounds. The efficacy of stem cells in promoting burn wound healing has been reviewed extensively [35,36]. Among the various types of stem cells, epidermal stem cells (ESCs) are particularly noteworthy for skin tissue regeneration due to their high proliferation rate, easy accessibility, and strong association with skin regeneration processes [37]. However, scientific researchers face challenges when culturing ESCs. These challenges include progressive aneuploidy, polyploidy, and the accumulation of mutations after several passages.

In contrast, mesenchymal stem cells (MSCs) share similar regenerative properties and can be isolated from a variety of tissues, including bone marrow, adipose tissue, peripheral blood, umbilical cords, Wharton’s jelly, and dental pulp [38]. MSCs were first identified in bone marrow in the 1970s [39]. These multipotent progenitor cells are derived from mesodermal origins. They possess self-renewal capabilities and can differentiate into various mesenchymal lineages, including osteoblasts, chondrocytes, myocytes, and adipocytes but not hematopoietic stem cells.

Among MSCs, bone marrow-derived MSCs (BMSCs) and adipose tissue-derived MSCs (ADMSCs) have been widely studied for their potential applications in wound healing. The ability to migrate to the wound bed and differentiate into skin fibroblasts has been demonstrated in BMSCs [40,41]. However, the isolation of BMSCs involves invasive and painful procedures. In contrast, ADMSCs reside in the hypodermal layer of skin. They are abundant and can be easily isolated from adipose tissue collected through liposuction. Therefore, ADMSC is an attractive source for cell transplantation therapies in regenerative medicine [42]. Moreover, they can also differentiate into fibroblasts to aid in tissue regeneration.

Despite the advantages of mesenchymal stem cells (MSCs), several limitations hinder their widespread clinical application. First, the freezing and thawing processes can significantly reduce the reliability of MSCs. These processes alter their survival rates and proliferation abilities by affecting the physicochemical and biophysical reactions involved in cryopreservation [43]. Second, thawing decreases the immunomodulatory and blood-regulating effects of cryopreserved MSCs. It leads to a more rapid complement-mediated elimination upon exposure to blood [44,45]. Furthermore, the poor survival of MSCs when administered to avascular wounds, the potential risk of tumorigenesis, and the unclear scope of their clinical applications also limit the application of stem cell therapies in wound healing [46,47].

### 7.4. Soluble Factors Derived by MSCs

Recent studies indicate that wound healing mediated by stem cells involves the release of extracellular matrix (ECM) molecules, cytokines, and growth factors. These components are crucial for activating fibroblast activity and promoting M2 macrophage polarization [48,49,50]. These soluble ECM factors can be collected through various methods, primarily as secretome or extracellular vesicles (EVs). Their therapeutic potential in wound healing has been summarized in the recent literature [51].

The secretome derived from the supernatant of MSC cultures has been shown to promote wound healing in full-thickness excision models through anti-inflammatory activity [52]. A study on burn wounds demonstrated that a conditioned medium from human amniotic MSCs healed injuries in mice effectively by inhibiting skin cell apoptosis and promoting proliferation via the activation of the PI3K/AKT signaling pathway [53]. Also, the secretome collected from 3D-cultured MSCs recently was found to be effective to stimulate angiogenesis in vitro and enhance burn wound healing in vivo [54].

EVs can be classified based on their biogenesis mechanism, concept, and size [55]. They can be categorized as exosome, microvesicle, apoptotic EV, autophagic EV, stressed EV, matrix vesicles, exomere, and non-vesicular particles. Exosomes are small EVs (40–150 nm) originating from the inward budding of the endosomal membrane, forming multivesicular bodies. They contain a variety of bioactive molecules, including proteins, lipids, and nucleic acids (such as mRNA and microRNA). Exosomes derived from MSCs, including adipose-derived, bone marrow-derived, placental (such as human amniotic and umbilical cord-derived), fetal dermal, menstrual blood stromal/stem cells, and even induced pluripotent stem cells, are commonly used in wound healing research [56]. Exosomes from human epidermal stem cells (ESCs-Exo) have been shown to promote excisional skin wound healing in mice, with mechanisms related to the stimulation of proliferation and migration of human skin fibroblasts [57]. Another study found that exosomes from keratinocytes derived from induced pluripotent stem cells accelerated healing in deep second-degree burn wounds in mice through the miR-762-mediated promotion of keratinocyte and endothelial cell migration [58].

Despite the promising effects of secretome and exosomes on burn wound healing, significant challenges remain before their clinical application. Current knowledge regarding their therapeutic dose, mode and frequency of administration and potential side effects is limited and requires further investigation. Extensive long-term follow-up clinical trials with large sample sizes across multiple centers are indispensable to monitor their chronic effects.

### 7.5. Nano-Technology

Nano-technology is a novel technology that promotes burn wound healing. Many nanomaterials are being used to deliver antimicrobials, growth factors, and other drugs to the wound site [59,60]. These nanomaterials include nanoparticles, nanofibers, nanogels, and nanoemulsions. A study has demonstrated the remarkable efficacy of biopolymeric mats loaded with herbal drugs and green-synthesized zinc oxide nanoparticles through both in vitro and in vivo experiments. These mats exhibit excellent biocompatibility. They promoted collagen deposition and provided significant antibacterial effects at the wound site [61]. Electrospun biodegradable nanofiber wound dressings also exhibit promoting effects on burn wound healing. The application of a poly (lactic-co-glycolic acid) nanofiber with propolis solution facilitated burn wound healing in a porcine model [62]. In addition, a chitosan/N,N,N-trimethyl chitosan nanofiber dressing exhibited strong antibacterial activity, high swelling capacity, and fluid exchange properties. It also increased the wound closure rate in a rat deep burn model [63].

Nano-technologies have a great capacity to promote burn wound healing. More studies are necessary in order to illustrate the underlying therapeutic mechanisms completely and verify safety issues for extensive clinical applications.

### 7.6. Modernized Traditional Medicine

According to the World Health Organization (WHO), traditional medicine is often referred to as “alternative” or “complementary” medicine. It focuses on the use of traditional therapies to promote health and to prevent, diagnose, enhance, or treat both physical and mental illnesses [64,65]. In some Asian and African countries, nearly 80% of the population relies on traditional medicine for their primary healthcare needs. Common natural substances such as aloe vera and honey are widely used for treating burn wounds [66,67,68,69]. Additionally, various other herbal materials and formulations have been reported as effective treatments for burn wound management [70,71,72,73]. For instance, Robacin (a traditional ointment composed of *Rosa damascena*, *Calendula officinalis*, and beeswax) has demonstrated superior healing properties. It has shown minimal scarring and faster wound contraction when compared to other treatments, such as silver sulfadiazine, aloe vera, and Rimojen (another traditional ointment) [74].

Furthermore, traditional medicine can be significantly enhanced through the integration of modern technologies. For example, a hydrogel that incorporates water extracts of four herbs has demonstrated bactericidal, reparative, and immunostimulatory effects in the management of burn wounds in a rabbit model [75]. In addition, a biopolymeric mat loaded with herbal drugs and nanoparticles has provided a dual drug–antimicrobial strategy in wound dressing applications [61]. Bixin-loaded electrospun nanofibers have also been reported to increase the wound closure rate and reduce scar formation in a diabetic mouse model [76].

Traditional Chinese Medicine (TCM), with a history spanning over 2000 years, has played a crucial role in the Chinese healthcare system. The earliest known written description of burn wound treatment dates back to the work of Hong Ge (AD 281–341) [77]. Today, the principles of TCM are increasingly validated through modern scientific research, revealing its effectiveness on a global scale.

One notable TCM treatment for burn wounds is the Moist Exposed Burn Ointment (MEBO), also referred to as Moist Exposed Burn Therapy (MEBT). Developed in Beijing in 1989, MEBO is an oil-based herbal paste containing six herbal extracts with beta-sitosterol as the active ingredient. All of the ingredients are combined in a base of beeswax and sesame oil [78]. MEBO has been extensively studied, not only in China but also internationally. It has demonstrated effects, such as promoting angiogenesis, reducing fibrosis, inhibiting inflammation, and enhancing wound healing, in numerous animal studies [79,80] and clinical reports [81,82,83]. In addition, a clinical study found that a TCM powder composed of more than ten herbs effectively promoted burn wound healing recently [84]. The authors attributed this effectiveness to its ability to activate blood circulation, resolve blood stasis, eliminate pus, as well as facilitate new skin growth and muscle regeneration.

TCM can also be integrated with modern technologies to promote wound healing. The topical application of Danggui Buxue decoction-loaded liposomes in thermosensitive gel showed faster wound closure on dorsal full-thickness excisional wounds in rats [85]. A study of fibrous rhein (a major active ingredient of the Chinese herbal medicine Rhei Radix et Rhizoma) hydrogel demonstrated that it reduced infection and promoted wound healing on a mice burn wound model followed by infection with *S. aureus* [86]. In additional, the effect of nanofibers loaded with Astragaloside IV (an active ingredient in a Chinese medicinal herb Astragali Radix) on wound healing has been reported. One study illustrated that the Astragaloside IV nanofiber could significantly promote angiogenesis, increase the immune function of the wound, and inhibit scar formation [87]. Another study showed it promoted fibroblast proliferation, adhesion, and migration in vitro, and it accelerated wound healing in diabetic rats in vivo [88].

With the rapid development of biotechnology, traditional medicine can also play a significant role in advancing methods for facilitating burn wound healing.

## 8. Barriers in Translating Innovations for Facilitation of Burn Wound Healing

Despite the numerous innovative bench studies aimed at enhancing burn wound healing, their clinical application remains limited. Clearly, a significant gap exists between bench research and clinical practice in burn wound treatment methods. This gap can be attributed to several key limitations:

### 8.1. Variability in Animal Models

Many preclinical studies utilize animal models that may not accurately mimic human physiology, including differences in skin structure, healing processes, and immune responses. For example, the healing mechanisms in rodents can differ significantly from those in humans. The results, therefore, cannot be applicable to human patients directly. This variability can impair promising treatments that showed effectiveness in animal models but failed during clinical trials because they do not produce the same effects in humans.

### 8.2. Complexity of Burn Wounds

Burn injuries are not homogeneous. They can vary widely in terms of depth (e.g., first-degree, second-degree, third-degree burn), size, and etiology (e.g., thermal, chemical, electrical). This diversity complicates the development of universal treatments. A method that works well for one type of burn may be ineffective or harmful for another. Therefore, tailored approaches are often necessary, but they may slow down the progress of research aimed at finding broadly applicable solutions.

### 8.3. Lack of Standardization

The absence of standardized protocols for testing burn wound treatments can lead to discrepancies in study outcomes. Different research groups may use varying methodologies, dosages, and evaluation criteria. This situation makes it difficult to compare results across studies [89]. This lack of consistency can hinder the establishment of clear guidelines for clinical application and impede the adoption of effective treatments.

### 8.4. Limited Clinical Trials

A shortage of well-designed clinical trials frequently hinders the rigorous evaluation of the efficacy and safety of new burn wound treatments. Many clinical studies are observational or lack control groups. They gave weak evidence for the clinical use of emerging therapies. In the absence of robust trial data, clinicians may hesitate to adopt new treatment modalities. For safety, they prefer to rely on established conventional methods that may be less effective.

### 8.5. Interdisciplinary Collaboration

Bridging the gap between research and clinical practice often depends on collaboration among researchers, clinicians, and industry representatives. However, coordinating across these diverse fields is challenging. Variations in priorities, terminology, and working methods can create barriers for effective communication and teamwork. Consequently, it slows the translation of research into clinical practice.

### 8.6. Others

Additional factors, for instance, individual differences in patient responses to treatments that complicate the creation of one-size-fits-all solutions, lack of funding and resources to support comprehensive studies, and the lengthy and complex approval processes imposed by regulatory agencies, also contribute to the slow translation of research into clinical practice.

By addressing these limitations, the translation of burn wound treatment methods from bench to bedside can be improved.

## 9. Discussion and Conclusions

Burn injuries are among the most painful and debilitating forms of trauma, often affecting not only the skin but also deeper tissues such as muscles and blood vessels. These injuries typically require complex, multidisciplinary, and prolonged treatment approaches in order to manage pain, prevent infection, and promote tissue regeneration. Although conventional treatments, for instance, antimicrobial dressings, surgical debridement, and fluid resuscitation, have proven effective in reducing infection rates and lowering the mortality associated with severe burns, they often fail to achieve rapid and complete wound healing. Consequently, many patients are left with chronic wounds, scarring, and long-term functional impairments.

In recent years, remarkable progress in the fields of biotechnology and biomaterials has catalyzed the development of more advanced and effective burn treatment options. Among these innovations, modern hydrogel dressings can be tailored to deliver analgesic and anti-inflammatory agents directly to the wound site. They offer both pain relief and improved healing outcomes [20,21,22,23]. Additionally, emerging biological therapies, such as platelet-rich plasma (PRP), stem cells, secretomes, and extracellular vesicles (EVs), have shown promising regenerative potential in preclinical studies [31,35,36,37,40,41,42,51,52,53,54,56,57]. The advancements in nanomaterials, such as electrospun nanoparticles and nanofibers, also lead to a state-of-the-art treatment modality to reduce infection and to promote burn wound healing [59,60,61,62,63]. These novel interventions may further integrate with applications of modernized traditional medicines, which are being re-evaluated through contemporary scientific methods [61,76,79,80,81,82,84,85,86,87,88].

Despite the promising results from in vitro and animal studies, the clinical translation of the advanced therapies to facilitate burn wound healing remains limited. A key obstacle is the lack of well-conducted clinical trials, which are essential to establish the safety, efficacy, and reproducibility of these treatments in human populations. Without sufficient clinical evidence, integrating these innovative solutions into standard burn care protocols becomes challenging.

To accelerate the transition from research to clinical practice, it is critical to develop standardized and rigorously defined manufacturing processes for these therapies. Consistent procedures can ensure quality, scalability, and compliance with regulatory requirements. Additionally, researchers should actively collaborate with industry partners. The latter can provide the technical expertise, financial resources, and logistical support needed for large-scale clinical trials. Such partnerships can facilitate the design and implementation of multi-center clinical trials and, therefore, strengthen the evidence, which is necessary for regulatory approval.

Simultaneously, governments and regulatory bodies should help streamline approval pathways for novel burn therapies. Simplified and transparent regulatory frameworks can significantly reduce the time and cost of bringing innovative treatments to the market.

In conclusion, despite advancements in burn care, conventional treatments often fall short in promoting wound healing. Emerging biotechnological and biomaterial-based therapies show significant potential, but their clinical translation remains limited due to insufficient clinical trials and regulatory challenges. Standardized manufacturing, collaborative research, and streamlined approval procedures are essential to integrate these innovations into routine clinical practice and enhance burn care outcomes.

## Figures and Tables

**Figure 1 bioengineering-12-00428-f001:**
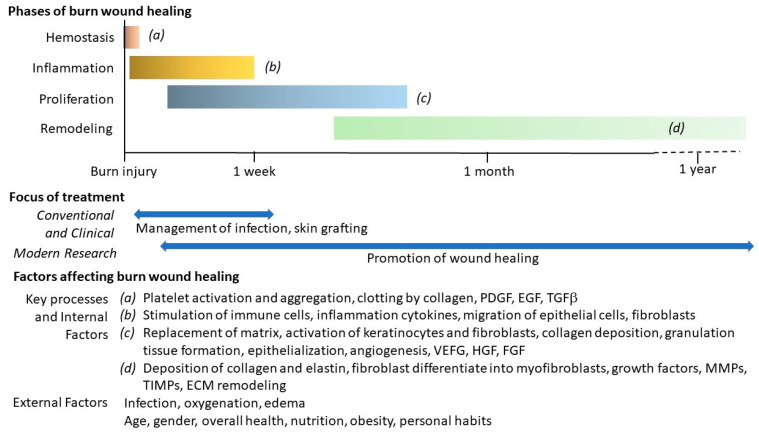
Summary of conventional treatments and modern research across the phases of burn wound healing and the factors affecting burn wound healing.

**Table 1 bioengineering-12-00428-t001:** Classification of burn wounds.

Burn Area	Burn Depth
Minor: <10% of TBSA in adults <5% in children and the elderly Primarily consist of superficial burns <2% classified as full-thickness burns	First-Degree Burns (Superficial Thickness): Affect only the uppermost layer The skin appears red, congested, and dry Mild pain
Major: >20% of TBSA in adults >10% in children and the elderly >5% as full-thickness burns Significant burns to critical areas such as the face, eyes, ears, joints, or genitalia	Second-Degree Burns (Partial Thickness): Partial damage to the dermis 2A: (Superficial Partial Thickness) Affect epidermis and the upper half of the dermis Blisters and weeping, painful Require dressing and wound care but typically do not need surgery 2B (Deep Partial Thickness) Affect deeper layers of the dermis More blisters, drier, result in more scarring, less painful Surgical intervention required
	Third-Degree Burns (Full Thickness): Damage the entire dermis, not painful Require protection against infection and surgical management
	Fourth-Degree Burns: Affect deeper tissues, including muscle and bone Burned areas blackened Significant loss of tissue

## Data Availability

No new data were created or analyzed in this study. Data sharing is not applicable to this article.

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
