# Peer review of "Review on Current Advancements in Facilitation of Burn Wound Healing"

_bioengineering, 2025, doi:10.3390/bioengineering12040428_

Round 1
Reviewer 1 Report
Comments and Suggestions for Authors
2025-03-18
The review of the submission bioengineering-3553006, entitled “Review on Current Advancements in Facilitation of Burn Wound Healing”.
First of all, let me acknowledge that the text is filled with the very strange statements, like:
1. “Burns are common injuries, but treating burn wounds can be complex.”
I am surprised, that authors parallel frequency of wound types with the complexity of their treatment.
2. “The review concludes that large scale clinical trials are essential to expedite the translation of these innovations into effective treatments for burn wound healing”
Indeed, what a discovery!
As a matter of fact, even taking into account that this is a review, I haven’t found any fresh thought.
Authors need to carefully underline the novelty and reason to be published
Major revisions
Comments on the Quality of English Language2025-03-18
The review of the submission bioengineering-3553006, entitled “Review on Current Advancements in Facilitation of Burn Wound Healing”.
First of all, let me acknowledge that the text is filled with the very strange statements, like:
1. “Burns are common injuries, but treating burn wounds can be complex.”
I am surprised, that authors parallel frequency of wound types with the complexity of their treatment.
2. “The review concludes that large scale clinical trials are essential to expedite the translation of these innovations into effective treatments for burn wound healing”
Indeed, what a discovery!
As a matter of fact, even taking into account that this is a review, I haven’t found any fresh thought.
Author Response
The review of the submission bioengineering-3553006, entitled “Review on Current Advancements in Facilitation of Burn Wound Healing”.
First of all, let me acknowledge that the text is filled with the very strange statements, like:
- “Burns are common injuries, but treating burn wounds can be complex.”
I am surprised, that authors parallel frequency of wound types with the complexity of their treatment.
Responses:
It presents two related ideas: that burns are frequently encountered as injuries and that their treatment may involve complexities, such as the severity of the burn, the area affected, and potential complications. This contrast between the commonality of the injury and the intricacies of its treatment.
However, after considering your comment, the sentence has been revised by a native English speaker.
- “The review concludes that large scale clinical trials are essential to expedite the translation of these innovations into effective treatments for burn wound healing”
Indeed, what a discovery!
Responses:
Thank you for the comment. This conclusion has been revised after an extensive revision of the manuscript.
As a matter of fact, even taking into account that this is a review, I haven’t found any fresh thought.
Authors need to carefully underline the novelty and reason to be published
Responses:
Thank you for your comment. The discussion on the integration of modern technique and traditional medicine to treat burn wound has been strengthened. Also, the barriers of translation of modern research to their clinical application has also augmented. These may give new insights to the readers.
Major revisions
Reviewer 2 Report
Comments and Suggestions for Authors
Burn wound treatment is a major public health issue. This review describes approaches to promote burn wound healing. Dwelling on the latest scientific developments designed to "facilitate" the healing of burns at each stage. And also the problems associated with the complexity of burn wounds, the lack of standardization in testing and the absence of adequate animal models, which inevitably leads to the need for further research carried out jointly by doctors, molecular biologists and physiologists.
The paper is written in a simple and understandable language and clearly presents the current state of the art in wound care, the direction of scientific development of new approaches to treatment and the problems that researchers face.
Dear authors, There are only a few comments on the paper.
The text, starting with line 73, should be separated by a separate heading, since it does not contain information on the classification of burns.
Line 255-256 contains an outdated classification of extracellular vesicles. The reference, which is given at the end of the sentence, should be replaced with an appropriate statement. Although the classification of exosomes is not crucial for this work, the classification and references should be up-to-date.
There is apparently a typo on line 256. A reference has been inserted that breaks up a word.
Line 257 According to the prescription of the International Society of Extracellular Vesicles, the size of exosomes is 30-100 nm. If we are talking about extracellular vesicles in general, their size ranges from 30 nm to 1 μm. I cannot agree that exosomes are organelles. Multivesicular bodies containing intraluminal vesicles, which after excretion from the cell are called exosomes, are certainly organelles, but not exosomes themselves.
What is the difference between secretome samples and exosome samples? And how can they be technically separated if they have the same source?
Author Response
Burn wound treatment is a major public health issue. This review describes approaches to promote burn wound healing. Dwelling on the latest scientific developments designed to "facilitate" the healing of burns at each stage. And also the problems associated with the complexity of burn wounds, the lack of standardization in testing and the absence of adequate animal models, which inevitably leads to the need for further research carried out jointly by doctors, molecular biologists and physiologists.
The paper is written in a simple and understandable language and clearly presents the current state of the art in wound care, the direction of scientific development of new approaches to treatment and the problems that researchers face.
Dear authors, There are only a few comments on the paper.
The text, starting with line 73, should be separated by a separate heading, since it does not contain information on the classification of burns.
Responses:
Thank you very much for your comment. A separate heading “4. Different phases in burn wound healing” is added accordingly.
Line 255-256 contains an outdated classification of extracellular vesicles. The reference, which is given at the end of the sentence, should be replaced with an appropriate statement. Although the classification of exosomes is not crucial for this work, the classification and references should be up-to-date.
Responses:
Thank you for your comment. The classification of extracellular vesicles has been updated and the reference has been revised accordingly.
There is apparently a typo on line 256. A reference has been inserted that breaks up a word.
Responses:
Thank you for your indication. The typo has been corrected.
Line 257 According to the prescription of the International Society of Extracellular Vesicles, the size of exosomes is 30-100 nm. If we are talking about extracellular vesicles in general, their size ranges from 30 nm to 1 μm. I cannot agree that exosomes are organelles. Multivesicular bodies containing intraluminal vesicles, which after excretion from the cell are called exosomes, are certainly organelles, but not exosomes themselves.
Responses:
Thank you very much for your informatic comments. The concept of exosome has been clarified and the sentence has been revised.
What is the difference between secretome samples and exosome samples? And how can they be technically separated if they have the same source?
Responses:
By definition secretome encompasses all the molecules secreted by a cell or tissue, including proteins, lipids, nucleic acids, and other biomolecules such as cytokines, growth factors, enzymes, extracellular matrix components, and extracellular vesicles (including exosomes and microvesicles), while exosomes are a specific type of extracellular vesicle, typically 30-150 nm in diameter, containing specific proteins, lipids, and nucleic acids (like mRNA and microRNA).
Secretome can be collected from centrifuging the cell supernatant through a centrifugal filter unit for 45 minutes at a speed of 4000 rcf..
Exosomes are a subset of the secretome. The International Society for Extracellular Vesicles (ISEV) has published the latest Minimal Information for Studies of Extracellular Vesicles (MISEV) guidelines, which outline the criteria that should be followed for each preparation of EVs. Differential ultracentrifugation is the most commonly used technique for separating exosomes. However, other chemical-based techniques, such as density gradients, precipitation, and immune-isolation, are also available.
Reviewer 3 Report
Comments and Suggestions for Authors
The article titled “Review on Current Advancements in Facilitation of Burn Wound Healing”has been evaluated and requires significant revisions before it can be considered for publication.
1. The introduction section should be expanded to include a more comprehensive discussion on the burden of burn wound healing, along with relevant statistical data highlighting its impact on overall medical health. Additionally, it should emphasize the necessity of effective wound management strategies.
2. The final paragraph of the introduction should be relocated to the subsequent section that addresses classification based on severity by burn size.
3. The current manuscript predominantly features textual explanations; however, it would benefit from the inclusion of figures and tables. These visual elements will facilitate reader comprehension and provide a more detailed context for the subject matter.
4. It is advisable for the authors to create and incorporate a flow diagram that clearly illustrates the literature search methodology. This diagram should delineate the specific search terms employed, the databases or search engines utilized, and the exclusion criteria applied during the literature selection process.
5. The authors must incorporate recent advancements in techniques involving electrospun nanofibers and the development of nanoparticles for wound healing applications.
6. The conclusion section requires thorough enhancement, featuring an in-depth discussion of the findings, their implications, and their connections to existing literature on the topic.
7. The current manuscript exhibits a similarity index of 27%. This figure should be reduced to below 20% to comply with publication standards.
8. A comprehensive linguistic edit by native English speakers is strongly recommended to rectify grammatical errors and awkward phrasings in the manuscript. Such improvements will enhance the clarity, precision, readability, and overall professionalism of the language used.
Comments on the Quality of English Language
A comprehensive linguistic edit by native English speakers is strongly recommended to rectify grammatical errors and awkward phrasings in the manuscript. Such improvements will enhance the clarity, precision, readability, and overall professionalism of the language used.
Author Response
The article titled “Review on Current Advancements in Facilitation of Burn Wound Healing”has been evaluated and requires significant revisions before it can be considered for publication.
- The introduction section should be expanded to include a more comprehensive discussion on the burden of burn wound healing, along with relevant statistical data highlighting its impact on overall medical health. Additionally, it should emphasize the necessity of effective wound management strategies.
Responses:
Thank you very much for your comments. The introduction has been expanded and augmented accordingly.
- The final paragraph of the introduction should be relocated to the subsequent section that addresses classification based on severity by burn size.
Responses:
Thank you for your suggestion. The final paragraph of the introduction has been relocated accordingly.
- The current manuscript predominantly features textual explanations; however, it would benefit from the inclusion of figures and tables. These visual elements will facilitate reader comprehension and provide a more detailed context for the subject matter.
Responses:
Thank you for the comment. One table and one figure are added.
- It is advisable for the authors to create and incorporate a flow diagram that clearly illustrates the literature search methodology. This diagram should delineate the specific search terms employed, the databases or search engines utilized, and the exclusion criteria applied during the literature selection process.
Responses:
Thanks for the advice. The “Methods” section has been added. It documented literature search methodology that we used. It includes the search engines utilized, the search terms employed and the inclusion criteria.
- The authors must incorporate recent advancements in techniques involving electrospun nanofibers and the development of nanoparticles for wound healing applications.
Responses:
Thanks for the comment. Techniques involving nano-technologies for wound healing applications have been added.
- The conclusion section requires thorough enhancement, featuring an in-depth discussion of the findings, their implications, and their connections to existing literature on the topic.
Responses:
Thank you for your comment. The conclusion section has been rewritten according to your suggestion. The heading has been changed to “Discussion and Conclusion”.
- The current manuscript exhibits a similarity index of 27%. This figure should be reduced to below 20% to comply with publication standards.
Responses:
Thank you for your comment. The overall similarity index has been reduced to below 22% after the revision of the manuscript. From the report (VeriGuide), most of the similarities fall in Section 4 (Different phases in burn wound healing). In fact, the sentences in this section describe the general knowledge of the phases in burn wound healing. Therefore, it is the reason having a high similarity index. In fact, the wordings we used are different from those in other sources. In addition, this article is a review. One of its aims is to summarizes the idea and results from other literatures. This might be another reason to get a similarity index above 20%.
- A comprehensive linguistic edit by native English speakers is strongly recommended to rectify grammatical errors and awkward phrasings in the manuscript. Such improvements will enhance the clarity, precision, readability, and overall professionalism of the language used.
Responses:
Thank you for your suggestion. The whole manuscript has been reviewed and edited by a native English speaker.
Round 2
Reviewer 1 Report
Comments and Suggestions for Authors
It can be published in the current shape
Reviewer 3 Report
Comments and Suggestions for Authors
The authors made substantial revisions in response to the reviewer's comments, making the manuscript suitable for publication.